# Peat-Inhabiting *Verrucomicrobia* of the Order *Methylacidiphilales* Do Not Possess Methanotrophic Capabilities

**DOI:** 10.3390/microorganisms9122566

**Published:** 2021-12-11

**Authors:** Svetlana N. Dedysh, Alexey V. Beletsky, Anastasia A. Ivanova, Olga V. Danilova, Shahjahon Begmatov, Irina S. Kulichevskaya, Andrey V. Mardanov, Nikolai V. Ravin

**Affiliations:** 1Research Center of Biotechnology of the Russian Academy of Sciences, Winogradsky Institute of Microbiology, 119071 Moscow, Russia; ivanovastasja@gmail.com (A.A.I.); olga.v.danilova10@gmail.com (O.V.D.); kulich2@mail.ru (I.S.K.); 2Research Center of Biotechnology of the Russian Academy of Sciences, Institute of Bioengineering, 119071 Moscow, Russia; mortu@yandex.ru (A.V.B.); shabegmatov@gmail.com (S.B.); mardanov@biengi.ac.ru (A.V.M.); nravin@biengi.ac.ru (N.V.R.)

**Keywords:** acidic peatlands, *Verrucomicrobia*, *Methylacidiphilales*, microbial diversity, metagenomes, methanotrophic capabilities

## Abstract

Methanotrophic verrucomicrobia of the order *Methylacidiphilales* are known as extremely acidophilic, thermophilic or mesophilic bacteria that inhabit acidic geothermal ecosystems. The occurrence of verrucomicrobial methanotrophs in other types of acidic environments remains an open question. Notably, *Methylacidiphilales*-affiliated 16S rRNA gene sequences are commonly retrieved from acidic (pH 3.5–5.5) peatlands. In this study, we compared the patterns of verrucomicrobial diversity in four acidic raised bogs and six neutral fens located in European North Russia. *Methylacidiphilales*-like 16S rRNA gene reads displaying 83–86% similarity to 16S rRNA gene sequences of currently described verrucomicrobial methanotrophs were recovered exclusively from raised bogs. Laboratory incubation of peat samples with 10% methane for 3 weeks resulted in the pronounced increase of a relative abundance of alphaproteobacterial methanotrophs, while no response was detected for *Methylacidiphilales*-affiliated bacteria. Three metagenome-assembled genomes (MAGs) of peat-inhabiting *Methylacidiphilales* bacteria were reconstructed and examined for the presence of genes encoding methane monooxygenase enzymes and autotrophic carbon fixation pathways. None of these genomic determinants were detected in assembled MAGs. Metabolic reconstructions predicted a heterotrophic metabolism, with a potential to hydrolyze several plant-derived polysaccharides. As suggested by our analysis, peat-inhabiting representatives of the *Methylacidiphilales* are acidophilic aerobic heterotrophs, which comprise a sister family of the methanotrophic *Methylacidiphilaceae*.

## 1. Introduction

The occurrence of methanotrophic capabilities in members of the bacterial phylum *Verrucomicrobia* was discovered fifteen years ago in three independent studies that focused on exploring methanotrophic microbial communities of three geographically remote acidic geothermal sites located in New Zealand, Italy and Kamchatka [1,2,3]. By that time, all cultivated members of this phylum were neutrophilic heterotrophic bacteria with either aerobic or anaerobic lifestyles. In contrast, the three originally obtained isolates of methanotrophic verrucomicrobia, strains V4, SolV and Kam1, were represented by thermophilic and extremely acidophilic bacteria, which grew at 37–65 °C (optimum at 55–60 °C) in the pH range between 0.8 and 5.0 (optimum at pH 2–3.5). At the time of description, these isolates were given different provisional names but, based on the high similarity of their 16S rRNA gene sequences (>98.4%) they were later considered as belonging to one genus, *Methylacidiphilum* [4]. Later, three mesophilic species of methanotrophic verrucomicrobia were isolated from a volcanic soil in Italy and were assigned to the novel genus *Methylacidimicrobium* [5]. By now, more than a dozen strains have been isolated, belonging to the genera *Methylacidiphilum* and *Methylacidimicrobium*, which belong to the family *Methylacidiphilaceae* of the order *Methylacidiphilales*. None of these names were validly published but they are commonly cited in the literature without quotations.

The *Verrucomicrobia* methanotrophs are autotrophic bacteria, which assimilate carbon from carbon dioxide (CO_2_) using the Calvin-Benson-Bassham (CBB) cycle [5,6]. Methane (CH_4_) is used by these bacteria as energy source and is oxidized to methanol (CH_3_OH) by the particulate methane monooxygenase (pMMO) enzyme. The latter is encoded by the *pmoCAB* genes, which are divergent from those in proteobacterial methanotrophs [4]. Instead of calcium-dependent pyrroloquinoline quinone (PQQ) methanol dehydrogenase (Ca-MDH), which is present in all proteobacterial methanotrophs and is encoded by the genes *mxaFI*, the genomes of verrucomicrobial methanotrophs contain *xoxF* genes that encode an Ln-MDH with lanthanides instead of calcium in the active site [7]. Discovery of mixotrophy in these bacteria suggested that verrucomicrobial methanotrophs have evolved to simultaneously utilize hydrogen and methane to meet energy and carbon demand in habitats where nutrient flux is dynamic [8]. Extensive studies of verrucomicrobial methanotrophs over the past years revealed their high metabolic versatility (reviewed by Schmitz et al. [9]). These bacteria seem to be key players in multiple volcanic nutrient cycles and can utilize several gases and other compounds present in acidic geothermal ecosystems, such as methane, H_2_, CO_2_, N_2_, ammonium and perhaps also hydrogen sulfide [8,9].

The range of habitats colonized by the verrucomicrobial methanotrophs is not yet fully understood. These bacteria are commonly found in geothermal and volcanic environments in various parts of the world. *Methylacidimicrobium*-like 16S rRNA gene sequences have also been recovered from one man-made system, namely the biofilms in the crown of corroded concrete sewage pipes on Hawaii [10]. The occurrence of verrucomicrobial methanotrophs in other types of acidic environments has not been documented yet. Since *Methylacidiphilaceae* methanotrophs possess divergent *pmoA* genes, one of the very first studies on environmental distribution of these bacteria employed *pmoA*-based screening of samples from various acidic habitats, including acidic peatlands [11]. Notably, verrucomicrobial methanotrophs were found in several environments over a wide temperature range but were clearly restricted to geothermally influenced habitats. Yet, a number of other cultivation-independent studies reported the retrieval of *Methylacidophilales*-affiliated 16S rRNA gene sequences from acidic peatlands, mainly raised bogs [12,13,14]. These sequences, however, displayed only a low similarity (83–87%) to 16S rRNA gene sequences of currently described verrucomicrobial methanotrophs. The occurrence of the latter in acidic peatlands, therefore, remains an open question.

This study was initiated in order to get an insight in the biology of peat-inhabiting *Methylacidophilales* bacteria and to verify the presence of methanotrophic capabilities in these microorganisms by using incubation studies and metagenome analysis. In our previous comparative analysis of bacterial communities in two pairs of closely located acidic raised bogs and neutral fens we noticed that *Methylacidophilales*-affiliated 16S rRNA gene reads were recovered exclusively from raised bogs [14]. In this study, we verified and confirmed this observation using the extended pool of 16S rRNA gene reads which was retrieved from four raised bogs and six fens in our recent study [15]. Our further analyses were performed with peat samples collected from raised bogs. We did not detect any response of *Methylacidophilales* bacteria to methane availability as well as did not reveal MMO encoding genes in the corresponding metagenomes. Apparently, peat-inhabiting *Methylacidophilales* are aerobic chemoorganotrophic bacteria with some hydrolytic capabilities, which comprise a sister family of the methanotrophic *Methylacidiphilaceae.*

## 2. Materials and Methods

### 2.1. Analysis of Verrucomicrobia Diversity in Ten Different Peatlands of European North Russia

Our study began with the analysis of *Verrucomicrobia* diversity patterns in four raised bogs and six eutrophic fens located in the Vologda region of European North Russia, within the zone of middle taiga. For this purpose, we re-examined the 16S rRNA gene sequence dataset retrieved by Dedysh et al. [15] and deposited in Sequence Read Archive (SRA) under the accession numbers SRR11280489-SRR11280524 (Bioproject PRJNA610704). Detailed description of the sampling sites is reported elsewhere [14,15], while some key characteristics of these peatlands are provided in Table 1. The peat samples collected from bogs and fens differed in peat water pH and conductivity, total nitrogen contents as well as concentrations of Ca, Mg, Fe, and P. The above mentioned pool of 16S rRNA gene sequences was re-analyzed with *QIIME 2* v.2019.10 (https://qiime2.org) [16]. DADA*2* plugin was used for sequence quality control, denoising and chimera filtering [17]. Operational Taxonomic Units (OTUs) were clustered applying VSEARCH plugin [18] with open-reference function using Silva v. 132 database [19,20] with 97% identity. Taxonomy assignment was performed using BLAST against Silva v. 132 database with 80% identity.

### 2.2. Peat Sampling for Incubation Studies

To examine the response of peat-inhabiting representatives of the *Methylacidiphilales* to methane availability, one additional batch of peat samples was collected in September 2020 from two raised peat bogs, Shichengskoe (59°56′56″ N, 41°16′59″ E) and Piyavochnoe (60°46′29″ N, 36°49′35″ E). The samples were transported to the laboratory, homogenized and used for determination of CH_4_ oxidation activity, incubation experiments and molecular analyses.

### 2.3. Determination of CH_4_ Oxidation Activity of Peat Samples and Incubation Experiments

Weighted portions of peat (10 g) sampled from Piyavochnoe and Shichengskoe raised bogs were placed in sterile glass vials 160 mL in volume, which were then sealed hermetically. Methane was injected in the flasks up to the concentration of about 1000 ppm. The vials were incubated at room temperature for 24 h. Samples of the gas phase (0.5 mL) were taken from the flasks periodically and analyzed for methane concentration on a Kristall 5000 chromatograph (Khromatek, Yoshkar Ola, Russia). The experiments were made in triplicate. The rate of methane oxidation by the peat samples was calculated in mg CH_4_ g^−1^ of wet peat h^−1^. After these activity measurements, methane was added to the flasks up to the concentration of 10% (*v*/*v*) in the headspace. The flasks were then incubated at room temperature for 3 weeks. At the end of each week, the flasks were flushed with ambient air using a sterile filter (0.22 µm) to completely remove remaining CH_4_ and accumulated CO_2_, and methane was re-injected up to the concentration of 10% (*v*/*v*), in order to keep high methane availability during the whole incubation period. Portions of peat from each incubation flask were taken before and after incubation with CH_4_ and used for the molecular analysis.

### 2.4. 16S rRNA Gene Sequencing and Analysis

Total genomic DNA was isolated from peat taken before and after incubation with CH_4_ using a Power Soil DNA isolation kit (MO BIO Laboratories, Inc., Carlsbad, CA, USA) and stored at −20 °C.

PCR amplification of 16S rRNA gene fragments comprising the V3–V4 variable regions was performed using the universal primers 341F (5′-CCTAYG GGDBGCWSCAG) and 806R (5′-GGA CTA CNVGGG THTCTAAT) [21]. The obtained PCR fragments were bar-coded and sequenced on Illumina MiSeq (2 × 300 nt reads). Pairwise overlapping reads were merged using FLASH v.1.2.11 [22]. The final dataset consisted of 297,553 16S rRNA gene reads (16,824 to 34,406 reads per sample).

All sequences were clustered into operational taxonomic units (OTUs) at 97% identity using the USEARCH v. 11 program [23]. Low quality reads were removed prior to clustering, chimeric sequences and singletons were removed during clustering by the USEARCH algorithms. To calculate OTU abundances, all reads obtained for a given sample were mapped to OTU sequences at a 97% global identity threshold by USEARCH. The taxonomic assignment of OTUs was performed by searching against the SILVA v.138 rRNA sequence database using the VSEARCH v. 2.14.1 algorithm [18].

### 2.5. Sequencing of Metagenomic DNA and Assembly of MAGs

Total genomic DNA from 0.25 g of peat sample from Shichengskoe raised bog (BioSample SAMN14309750) was extracted using a Power Soil DNA isolation kit (MO BIO Laboratories, Inc.) and stored at −20 °C.

Metagenomic DNA was sequenced using the Illumina HiSeq2500 platform according to the manufacturer’s instructions (Illumina Inc., San Diego, CA, USA). The sequencing of a paired-end (2 × 150 bp) TruSeq DNA library generated 432,496,215 read pairs (~130 Gb). Adapter removal and trimming of low-quality sequences (Q < 30) were performed using Cutadapt v.3.4 [24] and Sickle v.1.33 (https://github.com/najoshi/sickle), respectively.

The resulting Illumina reads were de novo assembled into contigs using MEGAHIT v1.2.9 [25]. The obtained contigs were binned into metagenome-assembled genomes (MAGs) using MetaBAT v.2.15 [26]. The completeness of the MAGs and their possible contamination (redundancy) were estimated using CheckM v.1.1.3 [27] with lineage-specific marker genes. The assembled MAGs were taxonomically classified using the Genome Taxonomy Database Toolkit (GTDB-Tk) v.1.5.0 [28] and Genome Taxonomy database (GTDB) [29].

### 2.6. Genome Annotation and Analysis

Gene search and annotation of MAGs were performed using the RAST server 2.0 [30], followed by manual correction of the annotation by comparing the predicted protein sequences with the National Center for Biotechnology Information (NCBI) databases. The N-terminal signal peptides were predicted by Signal P v.5.0, and the presence of transmembrane helices was predicted by TMHMM v.2.0 (http://www.cbs.dtu.dk/services/TMHMM/). The average nucleotide identity (ANI) between the selected genomes was calculated using appropriate scripts from the Enveomics Collection [31]. GTDB-Tk v.1.5.0 was used to find single-copy marker genes in the MAGs and to construct a multiple sequence alignment of concatenated single-copy gene sequences, comprising of those from a given MAG and all species from the GTDB. A portion of the multiple sequence alignment generated in GTDB-Tk was used to construct a phylogenetic tree with PhyML v.3.3 [32] using default parameters. The level of support for internal branches was assessed using the Bayesian test in PhyML.

## 3. Results and Discussion

### 3.1. Verrucomicrobia Diversity Patterns in Peatlands

A total of 2,174,164 partial (average length ~440 bp) 16S rRNA gene sequences were retrieved from the examined peat samples. Of these, 1,024,783 sequences were retained after quality filtration, denoising and removing chimeras. The pool of *Verrucomicrobia*-affiliated reads included 85,526 sequences, which accounted for 12–15% and 4–7% of all bacterial reads retrieved from raised bogs and fens, respectively.

*Verrucomicrobia* community in the four raised bogs was dominated by members of the *Pedosphaeraceae* (from 49.6 ± 2.4% of total verrucomicrobial reads in Alekseevskoe bog to 55.9% ± 2.8% in Piyavochnoe bog, mean ± SE). Other major groups were the *Opitutaceae* (from 6.5% ± 1.1% in Barskoe bog to 24.0% ± 3.8% in Piyavochnoe bog) and *Chthoniobacteraceae* (from 8.0% ± 2.8% in Piyavochnoe bog to 25.0% ± 4.2% in Barskoe bog) (Figure 1). Less abundant groups of *Verrucomicrobia*, which were detected in all raised bogs, included *Methylacidiphilaceae* (from 6.7% ± 2.8% in Piyavochnoe bog to 9.2% ± 2.4% in Barskoe bog), *Puniceicoccaceae* (from 0.3% ± 0.3% in Alekseevskoe bog to 1.6% ± 0.6% in Barskoe bog) and S-BQ2-57 soil group (from 3.0% ± 1.7% in Shichengskoe bog to 7.4% ± 0.6% in Alekseevskoe bog) (Figure 1).

Similar to the raised bogs, the microbial assemblages in the six fens were dominated by members of *Pedosphaeraceae* (from 33.1 ± 5.7% of total verrucomicrobial reads in Ileksa fen to 53.8% ± 6.1% in Povreka fen), *Chthoniobacteraceae* (from 4.0% ± 1.2% in Rodionskoe to 40.1% ± 7.3% in Ileksa fen) and *Opitutaceae* (from 4.0% ± 1.5% in Charozerskoe to 39.0% ± 2.3% in Rodionskoe fen). Other numerically abundant groups were uncultured members of S-BQ2-57 soil group (from 2.3% ± 0.6% Rodionskoe fen to 17.7% ± 0.6% in Piyavochnoe fen), uncultured members of *Verrucomicrobiaceae* (from 1.0% ± 0.3% in Piyavochnoe to 6.2% ± 2.1% in Ileksa fen) and uncultured members of the order *Rubritaleaceae* (from 0.1% ± 0.1% in Rodionskoe to 6.0% ± 2.2% in Povreka fen). Interestingly, the latter group was absent from raised bogs. The representatives of *Chthoniobacterales* were detected only in Charozerskoe (1.1% ± 0.4%) and Shichengskoe fens (6.5% ± 0.7%) (Figure 1). Several minor groups of *Verrucomicrobia* that were detected both in bogs and fens included members of the *Terrimicrobiaceae*, *Xiphinematobacteraceae*, group DEV007 of *Verrucomicrobiales*, and uncultured representatives of *Chthoniobacterales* and *Verrucomicrobiae*. Thus, members of the *Methylacidiphilales* were present exclusively in raised bogs.

### 3.2. Most Abundant OTUs of Methylacidiphilales

Using 97% sequence identity, a total of 16 Methylacidiphilales-affiliated OTUs were identified in studied peat samples. Of these, four OTUs were detected in all raised bogs; four OTUs were unique for one of the four examined bogs and eight OTUs were shared between different bogs. The most abundant phylotype as well as several other OTUs displayed only a distant similarity (83–86%) to 16S rRNA gene sequences from described verrucomicrobial methanotrophs. The closest cultured phylogenetic relative (88–90% 16S rRNA gene sequence similarity) of peat-inhabiting verrucomicrobia was a rice-associated epiphytic bacterium, strain LW23 (genome accession number QAZA00000000) [33]. Highest sequence similarity (95–97%) was observed with environmental 16S rRNA gene clone sequences originated from Sphagnum mosses and various soils (acc. JN128688, HG529144, AM945464, JN128688, HG529024).

### 3.3. Methane-Induced Shifts in the Microbial Community Structure

To examine the response of peat-inhabiting *Methylacidiphilales* to methane availability, peat samples from two studied peatlands, Shichengskoe and Piyavochnoe raised bogs, were incubated with 10% (*v*/*v*) CH_4_ for 3 weeks. The bacterial community composition in these samples was analyzed before and after incubation with methane. The original methanotroph community in peat was dominated by representatives of the genus *Methylocystis* (Figure 2A), which comprised 75–95% of all detected methanotroph-related 16S rRNA gene reads. Minor groups of methanotrophs were represented by *Methylocapsa acidiphila*, *Methyloferula stellata* and members of the genera *Methylobacter* and *Methylomonas*. This pattern of methanotroph diversity is highly characteristic for acidic northern peatlands [34]. Incubation with methane resulted in a pronounced increase of methane-oxidizing activity of peat samples (Figure 2B). The response of methanotrophic community to methane availability in peat from Shichengskoe bog was stronger than that of methanotrophs colonizing Piyavochnoe raised bog.

The heatmap in Figure 3 illustrates the corresponding shifts in the bacterial community structure in peat from both raised bogs at the level of individual OTUs. The strongest response to methane availability was detected for *Methylocystis* spp., *Methyloferula stellata*, *Methylocapsa acidiphila*, and *Methylobacter* spp. No response was observed for the most abundant groups of peat-inhabiting heterotrophic bacteria. The only exception was represented by *Burkholderia*-affiliated OTUs 24 and 76, whose relative abundance peaked in response to incubation with methane. This response, however, was detected only in one peat sample from the raised bog Shichengskoe, while these bacteria were present in all other samples as well. No statistically significant responses of *Methylacidiphilales*-affiliated OTUs to CH_4_ availability were observed in this incubation experiment (Figure 3).

### 3.4. Assembly and Phylogenetic Placement of MAGs of Methylacidiphilales

To obtain MAGs of microbial community members, we sequenced the metagenome of peat sample from Shichengskoe raised bog using Illumina technique. According to 16S rRNA gene profiling, this microbial community comprised six OTUs assigned to the *Methylacidiphilales* with a total share of 1.1 % [14]. Analysis of the taxonomic affiliation of the obtained MAGs showed that three of them belong to the *Methylacidiphilales* (Table 2).

To analyze the phylogeny of the order *Methylacidiphilales,* we constructed a phylogenetic tree based on concatenated sequences of 120 conserved marker genes, including three *Methylacidiphilales* MAGs obtained in this work and 27 other *Methylacidiphilales* genomes, representing all species-level lineages recognized in the GTDB (Figure 4). All families and genera proposed by the GTDB within the *Methylacidiphilales* corresponded to well-separated monophyletic branches. This analysis placed all three MAGs obtained in this work and five MAGs available in the GTDB within the candidate genus JABDGH01 of the candidate family UBA1321 (Figure 4). The five GTDB MAGs were obtained from an alpine bog (GCA_903970275.1 and GCA_903970335.1), pond (GCA_903863295.1) and forest soil (GCA_013286145.1 and GCA_013285645.1). The pairwise ANI between all eight JABDGH01 genomes were between 76 and 80%, a value below the species boundary cutoff of 95% [35,36], suggesting that each of them represents a distinct species.

### 3.5. Genome-Based Metabolic Predictions

Among the three MAGs obtained, the SRB-291 genome was the most complete and, therefore, was analyzed in more detail. It was predicted to contain 3864 protein-coding genes and 49 tRNA genes. The most important metabolic hallmarks of autotrophic lifestyle of *Methylacidiphilaceae* methanotrophs is the presence of methane oxidation pathway, the Calvin cycle for autotrophic carbon fixation and the nitrogenase complex [5,6,9,37]. However, analysis of SRB-291 genome revealed no homologs of key genes for aerobic methanotrophy, namely, of particulate methane monooxygenase (*pmoCAB*). These genes were also missing in two other MAGs obtained in this work as well as in all other *Methylacidiphilales* genomes shown in Figure 4, except for *Methylacidiphilum, Methylacidimicrobium* and the candidate genus g_CADDYQ01 belonging to the family *Methylacidiphilaceae*. Likewise the presence of the Calvin cycle is limited to the *Methylacidiphilaceae*, as reveled by the absence of genes for two key enzymes, ribulose bisphosphate carboxylase (*rbcL* gene) and phosphoribulokinase in all other analysed *Methylacidiphilales* genomes. Nitrogenase genes were only found in the *Methylacidiphilum* and *Methylacidimicrobium* genomes (but are missing in MAG of the genus CADDYQ01), as well as in two other MAGs, Verrucomicrobia bacterium Tous-C9LFEB and Verrucomicrobium sp. I3b_bin-317 (Figure 4). The latter two enzymes however, were distant from *Methylacidiphilaceae* nitrogenases (<40% amino acid sequence identity) suggesting their different origin.

Analysis of the SRB-291 genome revealed genes encoding metabolic pathways common for aerobic heterotrophic bacteria, such as glycolysis, the tricarboxylic acid cycle, the pentose-phosphate pathway (both oxidative and non-oxidative parts) and an aerobic respiratory chain with the proton-translocating NADH-dehydrogenase complex, succinate dehydrogenase, quinol-oxidizing alternative complex III, and two cytochrome c oxidases of different types. The SRB-291 bacterium potentially has capacities for fermentative growth since its genome encodes enzymes for fermentative production of acetate from pyruvate, including pyruvate dehydrogenase and acetate kinase. The presence of NAD-coupled [NiFe] group 3 hydrogenase and formate dehydrogenase indicates that hydrogen and formate also could be generated as fermentation products. Pathways for anaerobic respiration were not found. Two other MAGs obtained in this study, SRB-18 and SRB-116, also encoded an aerobic respiratory chain indicating that these bacteria have similar metabolic properties.

The search for potential hydrolytic enzymes revealed that SRB-291 bacterium has a rather limited capacity to degrade complex organic substrates compared to specialized polysaccharide-degrading bacteria from *Sphagnum*-dominated peatlands, such as *Fimbriiglobus ruber* [38]. Nevertheless, the SRB-291 genome contains genes predicted to encode endo-1,4-β-xylanases of the GH10 and GH5 families, and the GH3 family hydrolase annotated as xylan 1,4-β-xylosidase or β-glucosidase. All these enzymes contained N-terminal secretion signal peptide suggesting their involvement in extracellular hydrolysis of xylan. The presence of intracellular enzymes of the isomerase pathway of xylose metabolism, xylose isomerase and xylulokinase, also suggests the ability of SRB-291 bacterium to grow on xylans.

The finding of beta-N-acetylglucosaminidase, an enzyme that cleaves N-acetyl-D-glucosamine (GlcNAc) from the non-reducing end of chitin oligomers [39], indicates that chito-oligosaccharides and GlcNAc could be used as a carbon and nitrogen sources. Upon import into the cytoplasm GlcNAc may be phosphorylated by N-acetylglucosamine kinase to yield GlcNAc-6-P. Then, GlcNAc-6-P could be deacetylated by N-acetylglucosamine-6-phosphate deacetylase yielding glucosamine 6-phosphate. The latter is converted via the action of glucosamine-6-phosphate deaminase into fructose-6-P that enters the Embden-Meyerhof pathway, while ammonium could be used as a nitrogen source.

An interesting feature of SRB-291 genome is the presence of the pathway for L-rhamnose metabolism [40]. The genome encodes N-terminal secretion signal- containing α-L-rhamnosidase that could perform extracellular hydrolysis of terminal non-reducing α-L-rhamnose residues in α-L-rhamnosides. Upon import into the cytoplasm, L-rhamnose could be converted to L-rhamnulose by L-rhamnose isomerase followed by its phosphorylation by rhamnulokinase. After that, bifunctional rhamnulose-1-phosphate aldolase/lactate dehydrogenase can cleave it unto dihydroxyacetone phosphate and lactaldehyde and convert the latter into lactyl-CoA. Lactate produced upon phosphorylation of lactyl-CoA could be converted to pyruvate by L-lactate dehydrogenase or membrane-associated Lut lactate dehydrogenase complex. Highly similar enzymatic pathway is used by bacteria for metabolism of L-fucose and the presence of L-fucose isomerase suggests that this sugar could serve as a growth substrate as well. The SRB-291 genome also harbors a large gene cluster encoding structural components of bacterial microcompartment (BMC), an organelle bounded by a proteinaceous shell. In heterotrophic bacteria, BMCs are usually used to concentrate and process volatile or toxic metabolites [41], such as lactaldehyde. Particularly, catabolic BMCs were shown to be involved in utilization of L-rhamnose in some planctomycetes, such as *Planctopirus limnophila* [42] and *Humisphaera borealis* [15].

As shown in our study, representatives of the order *Methylacidiphilales* are common inhabitants of acidic (pH 3.5–5.5) *Sphagnum*-dominated raised bogs. These bacteria, however, are absent from neutral (pH 6.5–7.6) eutrophic fens even though these two types of wetlands are located in a close proximity to each other. This suggests an obligate character of acidophily in peat-inhabiting representatives of the *Methylacidiphilales*. Analysis of the three MAGs reconstructed in our study allowed making an insight into the metabolic capabilities of these as-yet-uncultivated bacteria. As suggested by our analysis, peat-inhabiting *Methylacidiphilales* are aerobic heterotrophs, which have the potential to degrade xylan and to utilize a number of plant cell wall sugars, such as xylose and L-rhamnose. This range of potential growth substrates and the predicted presence of catabolic BMCs in cells suggest a notable lifestyle similarity of these verrucomicrobia to the recently described *Phycisphaerae* planctomycetes from wetlands [43]. Both groups of bacteria seem to be specialized in degrading plant-derived organic matter in peatlands. At the same time, methanotrophy can clearly be omitted from the list of potential metabolic capabilities of peat-inhabiting *Methylacidiphilales.* None of the genomic determinants characteristic for methanotrophic metabolism, such as the genes encoding methane oxidation enzymes and C_1_ assimilation pathways were detected in MAGs assembled for *Methylacidiphilales* from acidic wetlands.

As suggested by the phylogenomic analysis (Figure 4), *Methylacidiphilales* from raised peat bogs together with related bacteria from Alpine bogs, forest soils and some freshwater habitats comprise a sister family of the methanotrophic *Methylacidiphilaceae.* The only cultivated representative of this potential novel family is an epiphytic verrucomicrobium, strain LW23, which was characterized as being associated with the surface of rice roots [33]. A tentative name ‘*Astrimicrobium roseum*’ was proposed for this bacterium but its formal taxonomic description remains to be published. Strain LW23 was characterized as aerobic, neutrophilic to acidotolerant, mesophilic heterotrophic bacterium, which possesses a number of features potentially involved in plant-associated lifestyle. Notably, most MAGs of the candidate family UBA1321 were assembled from the samples composed of live and dead plant remains, such as the vegetation of Alpine bogs or poorly decomposed peat from upper peat bog layers. It may well be that many members of this verrucomicrobial family tend to associate with plants.

In summary, only the family *Methylacidiphilaceae* within the order *Methylacidiphilales* appears to host bacteria with methanotrophic capabilities, which apparently were acquired via the horizontal transfer of methane monooxygenase genes from an ancestral methanotroph [6] and further evolved as a group of bacteria specialized in utilization of gases and other compounds present in acidic geothermal ecosystems [8; 9]. All other members of the order *Methylacidiphilales*, most likely, are acidophilic or acid-tolerant heterotrophic bacteria with various substrate utilization preferences and environmental adaptations. Extending characterized diversity of these bacteria by using the information retrieved in metagenomic studies is one of the most promising strategies for further cultivation efforts.

## Figures and Tables

**Figure 1 microorganisms-09-02566-f001:**
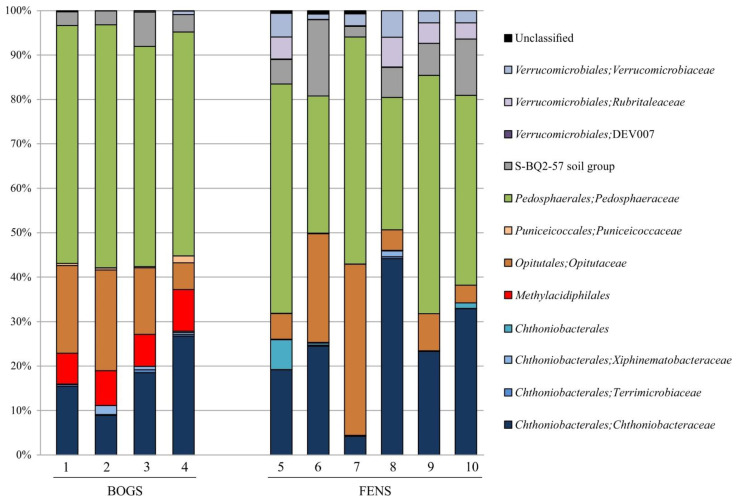
*Verrucomicrobia* community composition in raised bogs (1-Shichengskoe, 2- Piyavochnoe, 3- Alekseevskoe, 4- Barskoe) and fens (5-Shichengskoe, 6-Piyavochnoe, 7-Rodionskoe, 8-Ileksa, 9-Povreka, 10-Charozerskoe) according to the results of Illumina 16S rRNA gene sequencing. The composition is displayed at the order/family level. The relative abundance values represent averages of three replicate data sets. Taxa with the relative abundance of >0.1% at least for one ecosystem are shown.

**Figure 2 microorganisms-09-02566-f002:**
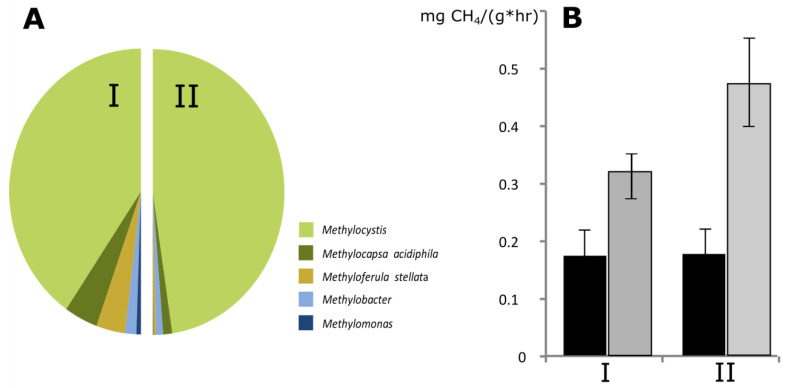
(**A**) Methanotroph community composition in peat samples from two raised bogs, Piyavochnoe (I) and Shichengskoe (II), as assessed by Illumina 16S rRNA gene sequencing. (**B**) Methane-oxidizing activity of peat samples from Piyavochnoe (I) and Shichengskoe (II) raised bogs before (black) and after (grey) incubation with 10% CH_4_ for 3 weeks.

**Figure 3 microorganisms-09-02566-f003:**
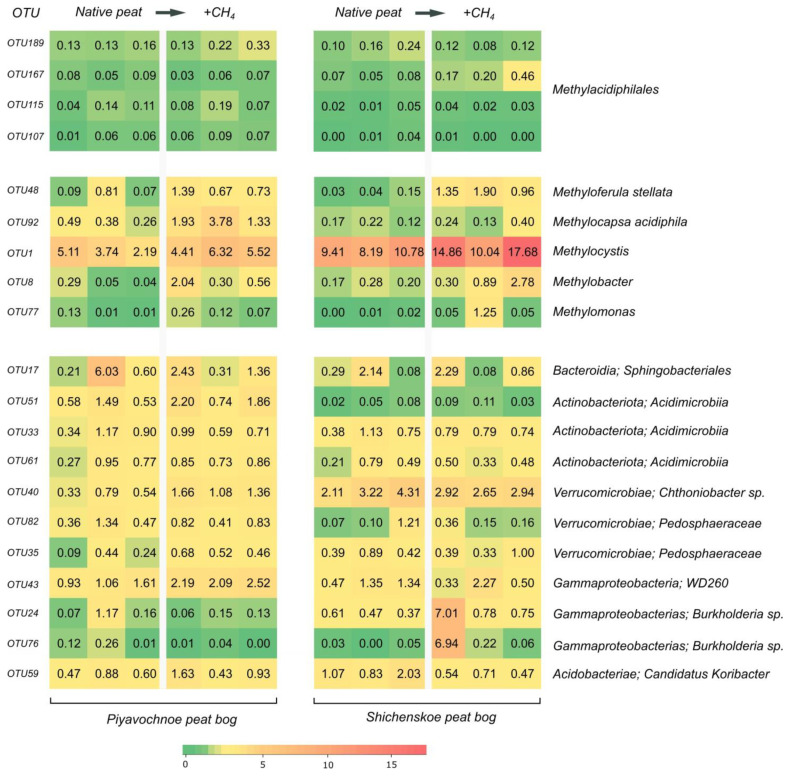
Heatmap showing the relative abundances of well-represented individual OTUs of *Methylacidiphilales*-affiliated bacteria, proteobacterial methanotrophs and heterotrophic bacteria in peat samples from Piyavochnoe and Shichengskoe raised bogs before and after incubation with 10% CH_4_ for 3 weeks. Graphs show three replicate data sets obtained for each group of samples.

**Figure 4 microorganisms-09-02566-f004:**
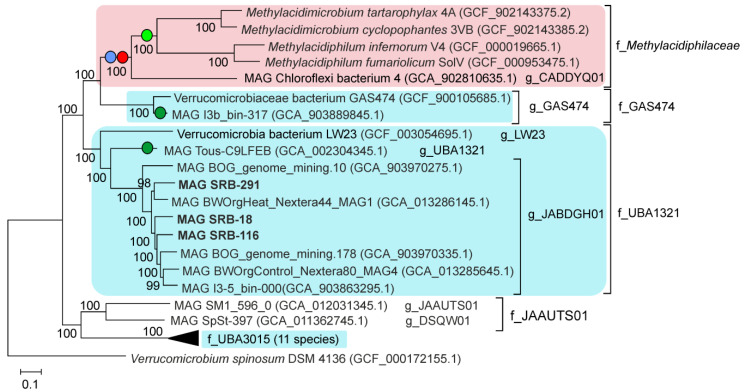
Genome-based phylogeny of the order *Methylacidiphilales* defined in the GTDB. Taxonomy is shown according to the GTDB (f_,family; g_, genus). The genome sequence of *Verrucomicrobium spinosum* DSM 4136 was used to root the tree. The presence of *pmoCAB* genes, the Calvin cycle and nitrogenase in particular lineages are indicated by red, blue and green circles, respectively. Light green and dark green circles indicate two different nitrogenase types. Lineages detected in geothermal habitats are marked in pink, while those from soil and aquatic environments are marked in blue.

**Table 1 microorganisms-09-02566-t001:** Characteristics of the peatlands examined in this study.

Mires		Characteristics								
		Coordinates	pH	Total Organic C (%)	N Total (%)	Sulfate (mg/L)	Fe (ppm)	Ca (ppm)	Mg (ppm)	P (ppm)
RAISED BOGS *	1	59°56′56″ N 41°16′59″ E	4.3	88.5	0.605	172	343	3522	634	614
	2	60°46′29″ N 36°49′35″ E	3.7	85.1	0.923	220	1347	4190	682	791
	3	59°27′10″ N 40°30′45″ E	4.3	88	0.685	211	662	4191	905	721
	4	59°22′33″ N 39°59′26″ E	4.1	81.5	1.16	200	5306	3765	816	1020
FENS **	1	59°56′31″ N 41°15′53″ E	7.4	73.6	2.31	202	9387	29,834	2575	1179
	2	60°46′08″ N 36°49′30″ E	6.9	71.6	1.65	222	16,344	27,373	1078	1305
	3	59°47′08″ N 37°52′08″ E	7.6	41.8	1.06	186	106,966	32,196	1599	8920
	4	61°08′18″ N 36°33′27″ E	6.9	83.2	2.55	230	3455	15,968	2583	1049
	5	61°07′16″ N 36°33′21″ E	6.5	48.6	1.51	607	19,264	8494	2665	1192
	6	60°30′42″ N 38°38′59″ E	7.1	66.2	2.4	188	5333	31,193	2695	985

* Raised bogs: (1) Shichengskoe, (2) Piyavochnoe, (3) Alekseevskoe, (4) Barskoe. ** Fens: Shichengskoe (1), Piyavochnoe (2), Radionskoe (3), Ileksa (4), Povreka (5), Charozerskoe (6).

**Table 2 microorganisms-09-02566-t002:** Main characteristics of *Methylacidiphilales*-affiliated MAGs obtained in this work.

MAG ID	MAG Size (bp)	Completeness (%)	Redundancy (%)	No. of Contigs	Sequencing Coverage	No. of tRNA Genes	No. of Protein-Coding Genes
SRB-18	2,871,672	90.41	3.04	88	19.7	35	2843
SRB-116	3,197,352	92.57	8.62	93	42.3	39	3241
SRB-291	3,829,391	95.11	5.6	169	9.2	49	3864

## Data Availability

The raw data generated from sequencing of the 16S rRNA gene fragments and metagenome have been deposited in the NCBI Sequence Read Archive (SRA) under the accession numbers SRR16962253-SRR16962264 and SRR16963054, respectively (BioProject PRJNA610704). The annotated sequences of three *Methylacidiphilales* MAGs have been deposited in the GenBank database and are accessible via the BioProject PRJNA610704 (under processing).

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
