# Peer review of "Peat-Inhabiting Verrucomicrobia of the Order Methylacidiphilales Do Not Possess Methanotrophic Capabilities"

_microorganisms, 2021, doi:10.3390/microorganisms9122566_

Round 1

Reviewer 1 Report

Comments and Suggestions for Authors

It is my immense pleasure to review this manuscript by Dedysh et Al., the work provides interesting findings and such results seem compatible with the special issue “Soil Microbiome: Biotic and Abiotic Interactions” the manuscript has not yet met the journal standard so the current manuscript can be accepted after these minor revisions.  

Novelty: the purpose of this manuscript is not clearly stated in this manuscript. Authors should describe the novelty compared to other similar published articles. (Mixotrophy drives niche expansion of verrucomicrobial methanotrophs | The ISME Journal (nature.com)

Grammatical errors:

A final proofread by a native English scientist is essential due to repetition of similar ideas in different paragraphs.

Line 134: no space between number and headings.

Materials and methods

Line 125-132: The method of gas injection is not clearly stated, please write down the complete protocol. Was gas completely removed from sterile tubes each week and replaced with new? What was the composition of injected sterile air?   

There is no detail about physicochemical properties of sampled soil. Will suggest to add some details like TC, TOC, pH etc.

There is no information about experimental design. Why author chose four acidic raised
bogs and six neutral fens for this study and there is no detail of sampling sites for each sample.

Results:

Results are well written and well presented.

Author Response

Comment: Novelty: the purpose of this manuscript is not clearly stated in this manuscript. Authors should describe the novelty compared to other similar published articles. (Mixotrophy drives niche expansion of verrucomicrobial methanotrophs | The ISME Journal (nature.com)

Response: The reference to the above mentioned study should have been included in the manuscript. This is our fault; we apologize. This reference is included now. The novelty of our study, however, cannot be compared to that of Carere and co-authors (2017) because these studies address two completely different groups of bacteria, methanotrophs and heterotrophs, which belong to two different families of the order Methylacidiphilales. In contrast to extensively studied verrucomicrobial methanotrophs, nearly nothing was known about their phylogenetic relatives that inhabit acidic peatlands. The purpose of our study is stated in the last paragraph of Introduction: “This study was initiated in order to get an insight in the biology of peat-inhabiting Methylacidophilales bacteria and to verify the presence of methanotrophic capabilities in these microorganisms by using incubation studies and metagenome analysis”.

Comment: A final proofread by a native English scientist is essential due to repetition of similar ideas in different paragraphs.

Response: One additional round of text proofreading was made in order to avoid repetition of similar statements. The text corrections we’ve made are highlighted by yellow.

Comment: Line 134: no space between number and headings.

Response: corrected.

Comment: Line 125-132: The method of gas injection is not clearly stated, please write down the complete protocol. Was gas completely removed from sterile tubes each week and replaced with new? What was the composition of injected sterile air?

Response: We have revised the corresponding text fragment and hope that it reads better now. The main purpose of flushing the experimental flasks each week was to avoid accumulation of CO2 and to keep high methane availability during the whole incubation period. We used ambient air and a sterile filter unit to replace gas in the flasks. 

Comment: There is no detail about physicochemical properties of sampled soil. Will suggest to add some details like TC, TOC, pH etc.

Response: The most important characteristics of the peatlands examined in our study are now provided in Table 1.

 Comment: There is no information about experimental design. Why author chose four acidic raised
bogs and six neutral fens for this study and there is no detail of sampling sites for each sample.

Response: No sampling was made in this study. The comparative analysis of verrucomicrobial diversity in peatlands was performed using the pool of 16S rRNA gene reads, which was retrieved from four raised bogs and six fens in our recently published study (Dedysh et al., 2021). We explain this at the very end of Introduction section as well as in Methods.

Reviewer 2 Report

The article by Svetlana N. Dedysh et al., “Peat-inhabiting Verrucomicrobia of the order Methylacidiphilales do not possess methanotrophic capabilities”,  get an insight in the biology of peat-inhabiting Methylacidophilales bacteria and reports on the absent of methanotrophic metabolisms of the verrucomicrobia members of this order. Thanks for the work clarifying that issue previously reported.

The manuscript is very well written, however I have some minor issues.

Minor comments:

Line 134: remove the dot between 16 and S.

Line 181, 182 and 184: missing “,” in the number of reads.

Line 214-215: Verrucomicrobiales should be italized

Fig 1 legend: - Verrucomicrobiae is not a class.

                     - Replace Verrucomicroiaceae by Verrucomicrobiaceae

                     - Puniceicoccaceae is not a family within  the order Opitutales

Line 360: replace Planctomyces limnophilus by Planctopirus limnophila

Author Response

Comment: Line 134: remove the dot between 16 and S.

Response: corrected.

Comment: Line 181, 182 and 184: missing “,” in the number of reads.

Response: corrected as recommended.

Comment: Line 214-215: Verrucomicrobiales should be italized

Response: corrected.

Comment: Fig 1 legend: - Verrucomicrobiae is not a class. Replace Verrucomicroiaceae by Verrucomicrobiaceae. Puniceicoccaceae is not a family within  the order Opitutales

Response: We have revised Figure 1 to fix these issues.

Comment: Line 360: replace Planctomyces limnophilus by Planctopirus limnophila

Response: Thank you for noticing this mistake. Done.